# ALIVE: Awakening LLM Reasoning
# via Adversarial Learning and Instructive Verbal Evaluation

## Abstract

The quest for expert-level reasoning in Large Language Models (LLMs) has been hampered by a persistent *reward bottleneck*: traditional reinforcement learning (RL) relies on scalar rewards that are **costly** to scale, **brittle** across domains, and **blind** to the underlying logic of a solution. This reliance on external, impoverished signals prevents models from developing a deep, self-contained understanding of reasoning principles. We introduce **ALIVE** (*Adversarial Learning with Instructive Verbal Evaluation*), a hands-free alignment framework that moves beyond scalar reward optimization toward intrinsic reasoning acquisition. Grounded in the principle of *Cognitive Synergy*, ALIVE unifies problem posing, solving, and judging within a single policy model to internalize the logic of correctness. By coupling adversarial learning with instructive verbal feedback, ALIVE enables models to internalize evaluative criteria directly from raw corpora, effectively transforming external critiques into an endogenous reasoning faculty. Empirical evaluations across mathematical reasoning, code generation, and general logical inference benchmarks demonstrate that ALIVE consistently mitigates reward signal limitations. With identical data and compute, it achieves accuracy gains, markedly improved cross-domain generalization, and higher self-correction rates. These results indicate that the reasoning trinity fosters a self-sustaining trajectory of capability growth, positioning ALIVE as a scalable foundation for general-purpose reasoning alignment without human-in-the-loop supervision.

[1]Anonymous Institution, Anonymous City, Anonymous Region, Anonymous Country. Correspondence to: Anonymous Author <anon.email@domain.com>.

Preliminary work. Under review by the International Conference on Machine Learning (ICML). Do not distribute.

## 1. Introduction

Large Language Models (LLMs) have achieved remarkable progress, evolving from mastering fundamental language understanding (Brown et al., 2020; Kojima et al., 2022) to complex reasoning tasks such as mathematics (Shao et al., 2024) and code generation (Hui et al., 2024). This success is driven not only by scaling pre-training data and model parameters (Kaplan et al., 2020; Hoffmann et al., 2022), but also by post-training techniques designed to improve reasoning, including reinforcement learning (RL) (DeepSeek-AI et al., 2025; OpenAI et al., 2024). These approaches allow models to self-correct and solve complex problems autonomously, unlocking emergent capabilities.

Despite these advances, RL-based reasoning is fundamentally limited by ***reward bottleneck***. Conventional RL relies on scalar reward signals that are **costly** to scale, **brittle** across domains, and **blind** to the underlying logic of a solution, preventing models from developing a deep, self-contained understanding of reasoning.

**1. Costly to scale.** RLHF (Ouyang et al., 2022) depends on expensive, noisy human feedback, while RLAIF (Lee et al., 2024) scales with synthetic rules but often introduces biases that misalign with real-world reasoning (Lambert, 2025). RLVR (Lambert et al., 2025; Jimenez et al., 2024) is restricted to easily constructed environments, limiting its use in realistic, open-ended tasks due to the high cost and complexity of building such settings.

**2. Brittle across domains.** Reward models typically exploit superficial correlations rather than fundamental reasoning (Wang et al., 2025b), requiring labor-intensive, task-specific reward engineering (Wang et al., 2024a; Mahan et al., 2024). Consequently, models trained in one domain rarely generalize to other domains, resulting in brittle performance across diverse tasks.

**3. Blind to the underlying logic of the reward.** Scalar or binary rewards discard the semantic structure of multi-step reasoning (Feng et al., 2024; Liu et al., 2025; Wang et al., 2025a; Lightman et al., 2023; Mukherjee et al., 2023; Luo et al., 2025), forcing models to rely on trial-and-error. Process-based rewards, chain-of-thought supervision, and verbal feedback help but still depend on curated data or

human oversight, leaving the reward bottleneck largely unresolved.

> *Can a language model autonomously construct reasoning tasks, solve them, and learn to evaluate its own reasoning directly from raw text, without relying on external reward annotation?*

We introduce ALIVE, a unified self-supervised reinforcement learning framework that integrates task construction, problem solving, and solution critique into a single policy. ALIVE forms a closed-loop adversarial process that internalizes reasoning correctness from raw text alone, eliminating the need for external reward annotation. As shown in Figure 1, ALIVE operates in three stages: **(1) Task Construction:** The model autonomously constructs reasoning tasks by masking valuable spans in raw text and generating corresponding ground-truth targets, creating a scalable and domain-agnostic source of supervision. **(2) Problem Solving:** The model produces complete reasoning trajectories and predicts the masked spans. **(3) Solution Review:** The model evaluates its own predictions using natural language critiques that explicitly assess correctness and reasoning quality. ALIVE circumvents the reward bottleneck by using self-generated, reasoning-rich verbal critiques derived from raw text as scalable, information-dense supervision.

Empirical evaluations across benchmarks for mathematical reasoning, code generation, and general logical inference demonstrate that ALIVE effectively alleviates the reward bottleneck. Under identical data and compute budgets, ALIVE delivers substantial performance gains, exhibits stronger cross-domain generalization, and achieves markedly higher self-correction rates. Collectively, these results indicate that the proposed reasoning trinity enables a self-reinforcing trajectory of capability growth, positioning ALIVE as a scalable foundation for general-purpose reasoning alignment that relies on no human-in-the-loop supervision.

Our main contributions are summarized as follows:

- We formally identify and systematize the **reward bottleneck** in RL-based reasoning, uncovering its fundamental limitations in scalability, robustness, and information efficiency.

- We introduce **ALIVE**, a unified self-supervised RL framework that enables LLMs to autonomously construct, solve, and review reasoning tasks directly from raw text.

- By leveraging self-generated rewards and instructive verbal feedback, ALIVE improves reasoning accuracy, cross-domain generalization, and intrinsic alignment without human-in-the-loop supervision.

- Through extensive experiments, we demonstrate that ALIVE consistently improves reasoning accuracy and

cross-domain generalization, establishing a scalable paradigm for intrinsic reasoning alignment without human-in-the-loop supervision.

## 2. Preliminaries

In this section, we formalize the **self-supervised reasoning** problem through the lens of *Cognitive Synergy*—the coordinated interaction of complementary cognitive roles within a single model to produce emergent reasoning capabilities. The ALIVE framework embodies this principle by unifying three roles—**Constructor**, **Solver**, and **Reviewer**—within a single language model $\pi_\theta$, enabling a self-contained learning cycle based solely on raw text.

### 2.1. Unified Cognitive Roles

We consider a large-scale unlabeled corpus $\mathcal{D} = \{d\}$. The goal is to train a single unified model $\pi_\theta$ that alternates between three roles, each with distinct input-output behavior:

**The Constructor.** Given a raw document $d$, $\pi_\theta$ first operates as a stochastic task generator. It performs $M$ independent rollouts to synthesize a diverse set of reasoning problems by identifying and masking different pivotal logical spans within the document. Each rollout $i \in \{1, \dots, M\}$ yields a query $\tilde{x}_i$ together with its corresponding *Hindsight Ground Truth* $y_i^*$:

$$\mathcal{T} = \{(\tilde{x}_i, y_i^*)\}_{i=1}^{M} \sim \pi_\theta(\text{construct} \mid d). \quad (1)$$

This process is deliberately adversarial: as training progresses, the model learns to construct masks that expose increasingly challenging reasoning gaps for itself.

**The Solver.** Given a constructed query $\tilde{x}_i$, $\pi_\theta$ performs stochastic rollouts to generate multiple candidate solutions by explicitly reasoning over the query. Each candidate output $\hat{y}_{ij}$ consists of a *reasoning trace* $z_{ij}$, which captures intermediate inference steps, followed by a *final answer* $a_{ij}$. The reasoning trace is treated as part of the model's generative output rather than an auxiliary annotation, allowing downstream evaluation to account for both the correctness of the answer and the coherence of the underlying reasoning process. We collect a group of $N$ such candidates via:

$$\mathcal{Y}_i = \{\hat{y}_{i1}, \dots, \hat{y}_{iN}\} \sim \pi_\theta(\text{solve} \mid \tilde{x}_i), \hat{y}_{ij} = (z_{ij}, a_{ij}). \quad (2)$$

**The Reviewer.** To assess candidate solutions, $\pi_\theta$ further instantiates a critic that evaluates each prediction in context. Specifically, it conditions on the triplet $(\tilde{x}_i, \hat{y}_{ij}, y_i^*)$, where $\hat{y}_{ij} = (z_{ij}, a_{ij})$ denotes a candidate reasoning–answer pair, and produces structured feedback:

$$(c_{ij}, v_{ij}) \sim \pi_\theta(\text{review} \mid \tilde{x}_i, \hat{y}_{ij}, y_i^*). \quad (3)$$

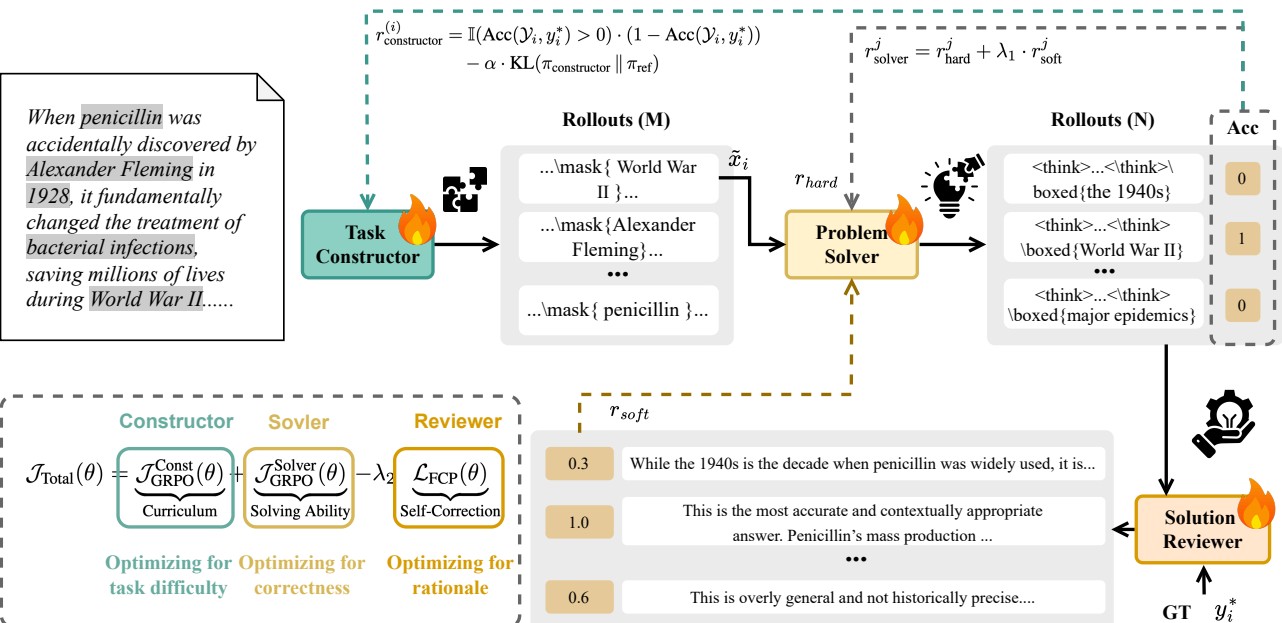

*Figure 1.* **Overview of the ALIVE framework.** A unified policy model $\pi_\theta$ alternates between three cognitive roles in a self-contained learning cycle: (1) **Constructor** masks critical spans in raw text to create tasks, (2) **Solver** generates reasoning trajectories for these tasks, and (3) **Reviewer** critiques its own solutions and provides both verbal feedback and soft rewards. The model parameters are updated by combining signals from all three roles, forming a closed-loop self-improvement system.

Here, $c_{ij}$ provides *Instructive Verbal Feedback* that articulates the reasoning behind success or failure, while $v_{ij} \in [0,1]$ serves as a *Pseudo Soft Reward* that quantifies partial logical correctness beyond binary accuracy.

### 2.2. Foundational Paradigms

Although the unified model $\pi_\theta$ shares parameters across all roles, it optimizes distinct objectives depending on its operational mode. The training signals are derived from the interactions between these roles:

**1. Adversarial Learning.** When operating as the Constructor, the Constructor is rewarded for masking valuable spans that minimize the *Solver*'s accuracy, strictly conditioned on a solvability constraint (accuracy $> 0$). The objective is to find masks that are challenging yet valid:

$$r_{\text{constructor}}^i = \mathbb{I}(\text{Acc}(\mathcal{Y}_i, y_i^*) > \epsilon) \cdot (1 - \text{Acc}(\mathcal{Y}_i, y_i^*)). \quad (4)$$

$$\text{Acc}(\mathcal{Y}_i, y_i^*) = \frac{1}{N} \sum_{j=1}^{N} \mathbb{I}(\text{ExactMatch}(\hat{y}_{i,j}, y_i^*)). \quad (5)$$

**2. Hybrid Rewarding.** When operating as the *Solver*, the model is optimized to maximize a hybrid scalar reward $r_{solver}^j(\hat{y}_{ij}, y_i^*)$, which combines two distinct signals: (1) **Verifiable Hard Reward ($r_{\text{hard}}$):** A sparse binary signal derived from exact matching against the hindsight ground truth $y^*$. (2) **Critical Soft Reward ($r_{\text{soft}}$):** A dense scalar

score $v_i$ generated by the model itself during the *Reviewer* phase, quantifying partial logical correctness.

**3. Verbal Feedback Learning.** Finally, to internalize the semantic information in the verbal critiques, we adopt the Feedback Conditional Policy (FCP) paradigm (Luo et al., 2025). Unlike standard RL, which compresses feedback into scalars, FCP treats the verbal critique $c$ as a **conditioning variable**. The model learns the posterior distribution $\pi_\theta(\hat{y}|\tilde{x}, c)$ by minimizing the negative log-likelihood on generated data pairs(especially negative):

$$\mathcal{L}_{\text{FCP}}(\theta) = -\mathbb{E}_{(\tilde{x}, \hat{y}, c)}[\log \pi_\theta(\hat{y}|\tilde{x}, c)]. \quad (6)$$

### 2.3. GRPO

We adopt Group Relative Policy Optimization(GRPO) (Shao et al., 2024) as the unified reinforcement learning backbone. The advantage function can be as follows:

$$A = \frac{r - \mu}{\sigma}, \quad (7)$$

where $\mu$ and $\sigma$ denote the mean and standard deviation of the group rewards. To ensure training stability, GRPO optimizes the policy by maximizing a surrogate objective within a trust region. Let $\pi_{\theta_{\text{old}}}$ denote the reference policy before the current update step. We define the probability ratio $\rho(\theta)$ for the generated trajectory as:

$$\rho(\theta) = \frac{\pi_\theta(y \mid x)}{\pi_{\theta_{\text{old}}}(y \mid x)}. \quad (8)$$

# 3. The ALIVE Framework

In this section, we present the **ALIVE** framework. Unlike traditional pipelines that rely on external reward models or static datasets, ALIVE orchestrates a self-contained, co-evolutionary game played by a single unified policy model ($\pi_\theta$). As illustrated in Figure 1, the framework iterates through three coupled phases: **Adversarial Task Synthesis**, **Introspective Evaluation**, and **Tri-Hybrid Optimization**. This cycle allows the model to autonomously discover, solve, and critique complex reasoning problems using only raw corpora.

## 3.1. Phase I: Adversarial Task Synthesis

ALIVE converts raw text into a dynamic curriculum of reasoning tasks through adversarial task construction. Following Xing et al. (2025), the model first assumes the role of a **Constructor**, which actively identifies and obscures pivotal logical spans within a document $d$. Given an input document, the Constructor generates a batch of $M$ distinct mask-span prediction tasks, formalized in Equation 1.

The Constructor is trained to *maximize task difficulty* by minimizing the Solver's prediction accuracy, thereby pushing generated tasks toward the Solver's decision boundary. However, aggressively masking context can lead to *unsolvable noise*, in which the task lacks sufficient information for meaningful reasoning. To prevent this failure mode, we introduce a *validity-gated reward mechanism* that explicitly enforces task solvability.

The reward for the $i$-th generated task is defined as:

$$r_{\text{constructor}}^i = \mathbb{I}(\text{Acc}(\mathcal{Y}_i, y_i^*) > 0) \cdot (1 - \text{Acc}(\mathcal{Y}_i, y_i^*))$$
$$- \alpha \cdot \text{KL}(\pi_{\text{constructor}} \| \pi_{\text{ref}}) \quad (9)$$

We design a validity-gated reward to synthesize tasks that are challenging yet solvable. The indicator $\mathbb{I}(\cdot)$ filters out unsolvable noise, ensuring the difficulty incentive $(1 - \text{Acc}(\mathcal{Y}_i, y_i^*))$ only activates for valid tasks. Combined with KL regularization to prevent policy degeneration, this mechanism drives the Constructor to autonomously explore the Solver's decision boundary:

$$\mathcal{J}_{\text{Const}}(\theta) = \mathbb{E}_{d \sim \mathcal{D}} \left[ \frac{1}{M} \sum_{i=1}^{M} \Big( \min \big( \rho_i \hat{A}_i, \right.$$

$$\left. \text{clip}(\rho_i, 1 - \epsilon, 1 + \epsilon) \hat{A}_i \big) - \alpha \, \mathbb{D}_{\text{KL}} \left( \pi_\theta(\cdot|d) \| \pi_{\text{ref}}(\cdot|d) \right) \Big) \right] \quad (10)$$

$$\hat{A}_i = \frac{r_{\text{constructor}}^i - \text{mean}(r_{\text{constructor}}^1, \cdots, r_{\text{constructor}}^M)}{\text{std}(r_{\text{constructor}}^1, \cdots, r_{\text{constructor}}^M)} \quad (11)$$

## 3.2. Phase II: Introspective Feedback Generation

Before detailing the optimization of the **Solver**, we first define the introspective evaluation mechanism. It generates the

**dense auxiliary signals** (verbal critiques and soft rewards) that constitute a critical part of the composite feedback driving the Solver.

In traditional RL pipelines, feedback is typically quantified into a scalar value to guide parameter updates. This process inevitably **compresses nuanced diagnostics into a scalar**, stripping away the semantic rationale required for effective error correction. ALIVE overcomes this *Blindness Bottleneck* by leveraging the **Reviewer** role. Since the task was constructed via masking, the model retains access to the *Hindsight Ground Truth $y^*$*. This allows it to act as a **White-Box** critic, diagnosing the reasoning logic before updates occur.

Specifically, for each composite output $\hat{y}_{ij} = (z_{ij}, a_{ij})$ generated by the Solver, the Reviewer utilizes the hindsight ground truth $y_i^*$ to perform a White-Box diagnosis. By scrutinizing the reasoning trace $z_{ij}$ and the final answer $a_{ij}$, the Reviewer synthesizes two distinct feedback signals:

**(1) Instructive Verbal Critique ($c_{ij}$):** The Reviewer diagnoses the reasoning trace $z_{ij}$ by conditioning on the final answer $a_{ij}$ and the ground truth $y_i^*$. This process explicitly articulates *why* the reasoning trajectory failed or succeeded, transforming implicit error signals into explicit natural language guidance:

$$c_{ij} \sim \pi_\theta(\text{critique} \mid y_{ij}, y_i^*) \quad (12)$$

**(2) Introspective Soft Reward ($r_{\text{soft}}$):** To mitigate the sparsity of binary outcome rewards, the Reviewer estimates a continuous quality score $v_{ij} \in [0, 1]$ derived from its evaluation. This quantifies partial logical correctness, allowing the optimization process to distinguish plausible reasoning attempts from complete hallucinations. Crucially, it also provides robustness against the potential non-uniqueness of constructed tasks, recognizing valid alternative solutions that diverge from the rigid hindsight ground truth.

## 3.3. Phase III: Optimization of the Reasoning Policy

In this phase, we focus on optimizing the **Solver** capabilities. To effectively utilize the hybrid feedback signals generated by the Reviewer (Phase II), we employ a dual-objective strategy that synergizes the goal-directed pressure of RL with the semantic guidance of verbal feedback.

We first construct a scalar reward $r_{solver}^j$ for each rollout. This involves the **Verifiable Hard Reward ($r_{\text{hard}^j}$)**, a strict binary signal derived from exact matching between the final answer $a_{ij}$ and the hindsight ground truth $y_i^*$:

$$r_{\text{hard}}^j = \mathbb{I}(\text{ExactMatch}(a_{ij}, y_i^*)). \quad (13)$$

We fuse this rigorous verification with the dense introspec-

tive soft reward ($r_{\text{soft}}^j$) generated by the Reviewer:

$$r_{\text{solver}}^j = r_{\text{hard}}^j + \lambda_1 \cdot r_{\text{soft}}^j \tag{14}$$

In this composite signal, $r_{\text{hard}}^j$ ensures convergence to the correct answer, while $r_{\text{soft}}^j$ provides dense gradients for intermediate reasoning steps.

**1. RL for Outcome Maximization.** To drive the Solver toward high-reward solutions, we adopt **Group Relative Policy Optimization (GRPO)** (Shao et al., 2024). We compute the advantage $A_{ij}$ by normalizing the composite reward $r_{\text{solver}}^j$ against the group mean.

Let $\rho_{ij}$ denote the probability ratio between the current and old policies for the generated trajectory $(z_{ij}, a_{ij})$:

$$\rho_{ij} = \frac{\pi_\theta(z_{ij}, a_{ij} \mid \tilde{x}_i)}{\pi_{\theta_{\text{old}}}(z_{ij}, a_{ij} \mid \tilde{x}_i)} \tag{15}$$

The policy is optimized to maximize the advantage within a trust region:

$$\mathcal{J}_{\text{Solver}}(\theta) = \mathbb{E}\left[ \frac{1}{N} \sum_{j=1}^N \left( \min\left(\rho_{ij} A_{ij}, \right.\right.\right.$$
$$\left.\left.\left. \text{clip}(\rho_{ij}, 1 - \epsilon, 1 + \epsilon) A_{ij}\right) - \beta \mathbb{D}_{\text{KL}} \right) \right], \tag{16}$$

$$A_{ij} = \frac{r_{\text{solver}}^j - \text{mean}(r_{\text{solver}}^1, \cdots, r_{\text{solver}}^N)}{\text{std}(r_{\text{solver}}^1, \cdots, r_{\text{solver}}^N)} \tag{17}$$

**2. Rationale Internalization.** To capture the semantic rationale lost by scalar rewards, we employ the **Feedback Conditional Policy (FCP)** (Luo et al., 2025). We treat the critique $c_{ij}$ as a *hindsight condition*, training the model to reconstruct its reasoning steps $\hat{y}_{ij}$ given the feedback. This is implemented as Maximum Likelihood Estimation (MLE), which formally minimizes the forward KL divergence to internalize the evaluation logic:

$$\mathcal{L}_{\text{FCP}}(\theta) = -\frac{1}{M \cdot N} \sum_{i=1}^M \sum_{j=1}^N \log \pi_\theta(z_{ij}, a_{ij} \mid \tilde{x}_i, c_{ij}). \tag{18}$$

This objective effectively transforms the Reviewer's critiques into verbal instructions, teaching the Solver the inverse dynamics of reasoning.

### 3.4. Phase IV: Global Unified Update

Since ALIVE operates as a single unified language model $\pi_\theta$ switching between roles, the final parameter update is a dynamic aggregation of signals from the entire self-play loop. As illustrated in Figure 1, the model is driven by **four distinct gradient sources**:

(1) **Task Difficulty Signal** (from Phase I): The *Constructor*'s reward $r_{\text{constructor}}$ encourages generating challenging yet solvable tasks.

(2) **Hard Verification Signal** (from Phase II): The binary outcome $r_{\text{hard}}$ enforces rigorous correctness.

(3) **Soft Introspective Signal** (from Phase II): The process score $r_{\text{soft}}$ rewards partial logical validity.

(4) **Verbal Diagnostic Signal** (from Phase II): The critique $c_{ij}$ provides semantic gradients via FCP.

Consequently, the total optimization objective for the unified parameters $\theta$ is formulated as a multi-task learning problem. We simultaneously maximize the expected returns for both the Constructor and Solver roles, while minimizing the negative log-likelihood for rational internalization:

$$\mathcal{J}_{\text{Total}}(\theta) = \underbrace{\mathcal{J}_{\text{Const}}(\theta)}_{\text{Curriculum}} + \underbrace{\mathcal{J}_{\text{Solver}}(\theta)}_{\text{Reasoning}} - \lambda_2 \underbrace{\mathcal{L}_{\text{FCP}}(\theta)}_{\text{Internalization}} \tag{19}$$

> **Takeaway**
>
> **ALIVE enables fully self-supervised reasoning learning at scale by closing the loop among task construction, reward generation, and feedback interpretation within a single model.**
>
> - **Scalable Self-Supervision from Raw Data.** Reasoning tasks are constructed directly from unlabeled text, removing reliance on curated datasets or human annotations.
>
> - **General, Self-Generated Rewards.** Both hard and soft rewards are generated internally by the model, eliminating the need for task-specific reward engineering and external reward models.
>
> - **High-Bandwidth Learning via Verbal Feedback.** Natural language critiques serve as conditioning signals, providing richer, more informative supervision than scalar rewards alone.

## 4. Experiments

In this section, we present a comprehensive empirical evaluation of ALIVE. We structure our experiments to answer three core questions:

**Q1: Can the framework learn effectively without costly external supervision?** We investigate whether ALIVE's self-supervised task construction and review pipeline can achieve strong reasoning performance while entirely eliminating the need for human or programmatic reward annotations (Section 4.2).

**Q2: Does the adversarial curriculum produce robust, domain-agnostic reasoning?** We examine whether the co-evolution of the Constructor and Solver leads to more robust reasoning across diverse tasks and domains, reducing the need for task-specific reward engineering (Section 4.3).

**Q3: Does verbal critique provide a richer learning signal than sparse rewards?** We assess whether fine-grained natural language feedback enables more efficient and stable learning than training with only binary outcome rewards (Section 4.4).

**Notes:** To isolate the contribution of the **Reviewer**—the only component not directly supervised by explicit ground-truth signals—we introduce two experimental configurations:

- **ALIVE-Self.** The standard setting in which the policy model $\pi_\theta$ generates the verbal critiques and soft rewards. Here, the Reviewer is trained exclusively through downstream Solver performance and the global unified objective, without access to external annotations.

- **ALIVE-Oracle.** A variant in which the verbal critique is generated by an advanced LLM, here we choose Kimi-K2 (Team et al., 2025a). This configuration provides an upper bound on diagnostic feedback quality, enabling a direct assessment of how closely the self-trained Reviewer approaches oracle-level guidance.

For reproducibility, training details, prompts, and further analysis are provided in Appendix C, F, and E.

## 4.1. Experiment Settings

**Critique Distillation Warm-up.** Base models initially lack instruction-following capability for the Reviewer role. We therefore initiate training with a 256-step warm-up phase where critique generation is distilled from a Teacher Oracle (Kimi-K2) (Team et al., 2025a). The model learns to produce diagnostic critiques $c_{\text{teacher}}$ conditioned on the triplet $(\tilde{x}, \hat{y}, y^*)$. Implementation specifics and coefficient schedules are detailed in Appendix D. In this phase, the Constructor generates $M = 8$ mask variations per raw document, and the Solver samples $N = 16$ solutions per mask, yielding 128 trajectories for teacher evaluation and distillation. The total effective input batch size per ALIVE loop is 265.

**Standard ALIVE Self-Play.** After warm-up, the model transitions to fully autonomous self-play, where $\pi_\theta$ acts as its own Reviewer. We maintain the same rollout configuration ($M = 8$, $N = 16$, 128 trajectories) but remove teacher supervision. The effective batch size thus reduces to 137 (1 document + 8 tasks + 128 FCP-style samples).

*Table 1.* Comparison of external feedback methods. ALIVE surpasses both scalar-RL (GRPO) and conditional-supervised (FCP) baselines, particularly on expert-level benchmarks like GPQA-Diamond.

| Method | General | | Mathematics | |
|---|---|---|---|---|
| | GPQA-D | MMLU-Pro | Math500 | AIME |
| Base Model | 27.9 | 49.7 | 63.8 | 7.5 |
| + GRPO | 32.5 | 49.7 | 75.7 | 20.0 |
| RFT | 35.2 | 55.0 | 69.2 | 13.3 |
| + GRPO | 37.2 | 57.0 | 75.1 | 25.8 |
| FCP | 35.0 | 53.6 | 68.7 | 7.5 |
| + Boostrap | 39.1 | 55.3 | 76.5 | 25.0 |
| **ALIVE-Self** | **45.96** | **61.32** | **78.4** | **26.67** |
| *ALIVE-Oracle* | *44.95* | *61.16* | *77.6* | *26.67* |

## 4.2. RQ1: Analysis of Scalability

We investigate whether ALIVE's self-supervised RL pipeline—which operates solely on raw text—can achieve strong reasoning performance, thereby eliminating the need for human or programmatic reward annotations. To evaluate its effectiveness, we compare ALIVE with the FCP (Luo et al., 2025) and external-reward RL methods (RFT and GRPO).

**Setup.** To ensure a rigorous comparison, we closely follow the experimental configuration of FCP (Luo et al., 2025). We adopt the same Qwen2.5-7B-Base model backbone and identical training corpora: Big-Math (Albalak et al., 2025), and WebInstruct (Ma et al., 2025). The key difference lies in how supervision is derived. For ALIVE, we first convert the original question-answer pairs into raw text documents $d = [x; y]$ using simple template concatenation. This unified format enables the Task Constructor to dynamically mask any informative span within the context, requiring the policy to reconstruct reasoning from partial information. Crucially, ALIVE operates in a fully self-supervised manner: it generates its own training signals and feedback directly from this raw text, without reliance on external reward models or human-provided corrections. In contrast, all baseline methods (FCP, RFT, GRPO) depend explicitly on the ground-truth answers or engineered reward functions from the same datasets as external supervision.

**Results and Analysis.** As shown in Table 1, ALIVE achieves superior performance compared to both external-reward RL (GRPO) and supervised feedback (FCP) baselines. This is especially pronounced on the GPQA-Diamond benchmark, where ALIVE attains **45.96%**, outperforming the strongest baseline (FCP+Bootstrap at 39.1%) by a substantial margin of over **6%** points. These results affirm that ALIVE's fully self-supervised pipeline is not only viable but highly effective, successfully learning strong reasoning capabilities without costly external reward signals. A key finding is that ALIVE-Self consistently matches or exceeds

the performance of ALIVE-Oracle, which utilizes ground-truth answers for verification. This indicates that the introspective review loop—where the model critiques its own reasoning—actively enhances learning, even in logic-dense domains like Big-Math and WebInstruct. The co-evolution of the Constructor, Solver, and Reviewer fosters a more unified and robust reasoning capability than static external supervision. This synergy validates the efficacy of the self-contained reasoning trinity framework, demonstrating its ability to form a robust, self-sustaining optimization cycle that effectively circumvents the traditional reward bottleneck.

### 4.3. RQ2: Analysis of Generalization

We examine whether the co-evolution of the Constructor and Solver within ALIVE's self-supervised loop leads to reasoning capabilities that generalize robustly across diverse and complex task domains. To this end, we evaluate ALIVE on two fronts: (1) long-horizon, open-ended agentic tasks that test procedural consistency, and (2) sensitivity to logical boundaries, which probes the model's intrinsic understanding of problem constraints rather than surface-level pattern matching.

**1. Long-Horizon Agentic Coding.** A core challenge for robust reasoning is maintaining logical consistency across extended sequences, especially in open-ended domains such as software engineering. We evaluate whether ALIVE's curriculum, built from adversarial task construction and dense verbal feedback, translates to stronger performance in Agentic Software Engineering. We utilize `Qwen3-30B-A3B-Instruct` (Yang et al., 2025) as the backbone, trained on a mixture of coding datasets (SWE-smith, CodeContests, NuminaMath), and evaluate on Live-CodeBench (Jain et al., 2024) and the more rigorous SWE-bench Verified (Jimenez et al., 2024).

*Table 2.* Performance on **Advanced Coding Benchmarks**. ALIVE demonstrates superior stability in multi-step agentic reasoning compared to scalar-only RL.

| Method (Qwen3-30B-Instruct) | LiveCodeBench Pass@1 | SWE-bench Ver. % Resolved |
|---|---|---|
| Base Model | 54.3 | 11.8 |
| SFT | 55.1 | 13.6 |
| GRPO (Scalar Reward) | 55.4 | 14.8 |
| FCP (Verbal Only) | 54.9 | 14.0 |
| **ALIVE-Self** | **56.0** | **17.2** |
| *ALIVE-Oracle* | *55.8* | *17.6* |

**Results and Analysis.** As shown in Table 2, ALIVE achieves **17.2%** on SWE-bench Verified, surpassing the scalar-reward GRPO baseline by a significant margin (+2.4%). This gain highlights the advantage of ALIVE's curriculum over static, outcome-only rewards. While FCP

provides verbal feedback, it lacks the adversarial pressure inherent in ALIVE's loop, making it less effective at handling the diverse edge cases present in SWE-bench. The results suggest that ALIVE's dynamic task construction and introspective review help the model internalize debugging logic—learning not just what code to write, but how to reason about potential failures in a complex, evolving context. This points to more robust, procedure-aware reasoning that generalizes beyond the patterns seen in static training data.

**2. Emergent Sensitivity to Logic Boundary.**

A robust reasoner must not only solve problems but also recognize when a problem is *unsolvable* due to missing information. Through its adversarial masking loop, ALIVE naturally encourages the model to identify logical pivots. To test if this creates an emergent awareness of logical sufficiency, we evaluate on **QuestBench** (Li et al., 2025), which benchmarks sensitivity to missing preconditions.

*Table 3.* Results on **QuestBench**. ALIVE significantly enhances the model's ability to detect logical gaps, particularly in planning tasks.

| Model | Logic-Q | Planning-Q |
|---|---|---|
| DeepSeek-V3.2 | 0.2713 | 0.2365 |
| GPT-4o | 0.3278 | 0.1451 |
| Kimi-K2 | 0.1513 | 0.2103 |
| Qwen3-30B-A3B-Instr | 0.4018 | 0.0850 |
| **ALIVE-Self (Qwen3-30B-A3B-Instr)** | **0.4391** | **0.3135** |

**Results and Analysis.** ALIVE training leads to a dramatic breakthrough in Planning-Q ($0.0850 \rightarrow 0.3135$), significantly surpassing both proprietary models like GPT-4o and DeepSeek-V3.2. We attribute this to the self-supervised reasoning trinity framework: the *Constructor*'s attempt to create difficult tasks by masking essential information forces the *Solver* to develop a "refusal" or "gap-detection" mechanism. This indicates that ALIVE moves beyond simple pattern matching toward an intrinsic understanding of logical completeness.

### 4.4. RQ3: Role of Verbal Feedback.

We investigate whether incorporating instructive verbal feedback enhances learning compared to **PretrainZero** (Xing et al., 2025). As the representative framework for adversarial active pretraining, PretrainZero relies exclusively on sparse outcome-based verification, serving as the ideal baseline to isolate the impact of dense verbal signals.

**Setup.** Our experiments follow the PretrainZero settings. We evaluate scalability across diverse model families: `SmolLM3-3B-Base` (Bakouch et al., 2025), `Qwen3-4B/8B-Base`, and `Qwen3-30B-MoE-Base` (Yang et al., 2025). All models share the same parameters $\pi_\theta$ and are trained for 2,048 steps on the Wikipedia corpus. More details are provided in Appendix C.3.

*Table 4.* Results on **General-Domain** reasoning benchmarks. Baseline results are cited from PretrainZero (Xing et al., 2025). ALIVE consistently outperforms the binary-reward baseline (PT-Zero) across all scales, proving that self-generated verbal diagnostics provide a denser, more effective learning signal than sparse binary outcomes.

| Method | SmolLM3-3B-Base | | | Qwen3-4B-Base | | | Qwen3-8B-Base | | | Qwen3-30B-MoE-Base | | |
|---|---|---|---|---|---|---|---|---|---|---|---|---|
| | MMLU-Pro | SuperGPQA | BBEH | MMLU-Pro | SuperGPQA | BBEH | MMLU-Pro | SuperGPQA | BBEH | MMLU-Pro | SuperGPQA | BBEH |
| Base | 16.66 | 12.62 | 3.32 | 51.94 | 26.32 | 8.67 | 59.19 | 31.12 | 10.49 | 58.79 | 33.73 | 10.51 |
| Rand. | 22.74 | 14.48 | 7.85 | 55.21 | 29.10 | 9.45 | 61.59 | 34.19 | 12.96 | 59.57 | 36.33 | 12.99 |
| PT-Zero | 32.41 | 19.44 | 3.78 | 60.37 | 32.28 | 12.68 | 64.28 | 34.46 | 14.67 | 64.59 | 36.58 | 14.91 |
| **ALIVE-Self** | **32.64** | 19.19 | **4.42** | **62.79** | **34.39** | **14.00** | **66.82** | **36.90** | **16.35** | **67.02** | **38.44** | **16.97** |
| *ALIVE-Oracle* | *34.85* | *21.10* | *5.07* | *63.15* | *34.82* | *14.56* | *66.92* | *36.85* | *16.39* | *67.15* | *39.20* | *16.79* |

*Table 5.* Results on **Math-Domain** reasoning benchmarks. On challenging tasks like AIME24, ALIVE demonstrates significant gains (e.g., **+4.06** on Qwen3-8B vs PT-Zero), verifying that verbal diagnosis is crucial when binary signals are sparse.

| Method | SmolLM3-3B-Base | | | Qwen3-4B-Base | | | Qwen3-8B-Base | | | Qwen3-30B-MoE-Base | | |
|---|---|---|---|---|---|---|---|---|---|---|---|---|
| | Math500 | GSM8K | AIME | Math500 | GSM8K | AIME | Math500 | GSM8K | AIME | Math500 | GSM8K | AIME |
| Base | 53.80 | 81.20 | 1.65 | 73.30 | 86.30 | 0.00 | 70.10 | 91.50 | 10.00 | 74.70 | 91.10 | 16.36 |
| Rand. | 59.00 | 82.50 | 0.00 | 74.80 | 87.50 | 10.00 | 79.20 | 93.80 | 13.30 | 79.20 | 82.40 | 14.58 |
| PT-Zero | 62.60 | 83.70 | 6.70 | 79.10 | 92.90 | 13.30 | 81.90 | 93.50 | 20.00 | 81.70 | 94.40 | 17.40 |
| **ALIVE-Self** | **64.00** | 83.40 | 6.35 | **83.80** | 92.72 | **13.44** | 83.80 | **96.06** | **24.06** | 83.00 | 95.53 | 19.69 |
| *ALIVE-Oracle* | *64.90* | *85.52* | *6.98* | *81.45* | *94.09* | *15.73* | *84.20* | *95.07* | *23.85* | *84.50* | *95.83* | *19.38* |

**Results and Analysis.** As shown in Tables 4 and 5, ALIVE consistently outperforms the binary-reward baseline (PretrainZero) across all model scales and domains, under identical training data and step budgets. Notably, on the challenging AIME24 benchmark, ALIVE with the Qwen3-8B model achieves a substantial gain of +4.06 points over PretrainZero.

These results provide direct empirical validation that **self-generated verbal feedback delivers a richer, more instructive learning signal than sparse binary rewards**. The performance lift is especially pronounced in complex reasoning tasks (e.g., AIME, SuperGPQA), where binary outcomes offer minimal guidance. Furthermore, the minimal gap between ALIVE-Self and ALIVE-Oracle—which uses ground-truth answers for critique conditioning—confirms that the model can effectively bootstrap its own diagnostic capability. This demonstrates that natural language critiques not only densify the reward signal but also enable stable, self-contained learning without external supervision.

## 5. Related Work

Recent research has increasingly focused on mitigating reliance on human annotation through self-evolving frameworks. Approaches like R-Zero (Huang et al., 2026) and PretrainZero (Xing et al., 2025) pioneer this direction by leveraging adversarial data generation, in which a generator dynamically generates challenging samples to bootstrap reasoning capabilities from scratch. Writing-Zero (Jia et al., 2025) further extends this dual-model gaming paradigm to non-verifiable tasks, bridging the gap between subjective generation and verifiable rewards. However, a critical limitation of these methods is their predominant dependence on sparse scalar rewards, which often fail to capture the fine-grained logic required for complex reasoning.

Complementarily, another line of work focuses on densifying the supervisory signal through generative verification. Research such as SCoRe (Kumar et al., 2024), Generative Verifiers (Zhang et al., 2025a), and Critique-GRPO (Zhang et al., 2025b) demonstrates that learning from self-generated corrections or instructive verbal critiques yields significantly superior performance compared to traditional scalar discriminators. To effectively utilize this rich information, methods like FCP (Luo et al., 2025) and NLAC (Hong et al., 2025) explicitly condition policies on such natural language feedback.

ALIVE synthesizes these two paradigms, unifying adversarial task construction with instructive verbal evaluation to establish a closed-loop system that autonomously scales task difficulty while internalizing deep reasoning signals.

## 6. Conclusion

This paper proposes ALIVE, a unified self-supervised reinforcement learning framework that addresses the *reward bottleneck* in reasoning tasks for LLMs. ALIVE integrates task construction, problem-solving, and self-review into a single policy, using self-generated verbal critiques rather than external rewards. Experiments across general-domain reasoning, mathematical problem solving, and agentic code generation demonstrate that ALIVE improves reasoning accuracy, cross-domain generalization, and emergent logical gap detection. This work establishes a scalable paradigm for intrinsic reasoning alignment without human-in-the-loop supervision.

## Impact Statement

This paper presents work whose goal is to advance the field of Machine Learning. There are many potential societal consequences of our work, none of which we feel must be specifically highlighted here.

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

## A. Related Work

Table 6. Comparison of training paradigms.

| Method | Annotation Free | Environment Free | Domain Free | Reward Signal | |
|---|---|---|---|---|---|
| | | | | Scalar | Verbal |
| RL(H/AI)F | ✗ | ✓ | ✗ | ✓ | ✗ |
| RLVR | ✓ | ✗ | ✗ | ✓ | ✗ |
| PretrainZero | ✓ | ✓ | ✓ | ✓ | ✗ |
| FCP | ✓ | ✓ | ✗ | ✗ | ✓ |
| **ALIVE (Ours)** | ✓ | ✓ | ✓ | ✓ | ✓ |

Existing approaches for enhancing reasoning in large language models vary in the types of supervision, environmental constraints, and reward signals they rely on. Methods such as RL(H/AI)F(Ouyang et al., 2022; Lee et al., 2024; Lambert, 2025) leverage human or AI feedback but require task-specific annotations and are often limited to a fixed environment and domain. RLVR(Wen et al., 2025) reduces annotation requirements but still relies on scalar rewards, providing little instructive feedback. PretrainZero(Xing et al., 2025) and FCP(Luo et al., 2025) improve generalization across domains or provide verbal guidance, yet each only partially addresses the limitations of prior paradigms—PretrainZero uses only scalar rewards, while FCP is restricted to certain domains.

In contrast, our proposed ALIVE framework unifies these dimensions: it is annotation-free, environment-agnostic, and domain-general, while simultaneously providing both scalar and verbal reward signals. As shown in Table 6, ALIVE effectively overcomes the key constraints of existing methods, enabling more efficient, flexible, and instructive self-supervised reasoning.

## B. Benchmark

To comprehensively evaluate model performance across diverse cognitive dimensions, we utilize a suite of benchmarks spanning general knowledge, mathematics, programming, and complex reasoning.

### B.1. General Domain

**MMLU-Pro.** MMLU-Pro (Wang et al., 2024b) is an enhanced version of the Massive Multitask Language Understanding (MMLU) benchmark, developed to address the saturation of performance on general knowledge tasks. It expands both the number of answer choices per question and the complexity of reasoning across 14 diverse subjects. By lowering the probability of success through random guessing, MMLU-Pro provides a more rigorous evaluation of a model's academic knowledge and reasoning abilities.

**GPQA-Diamond.** GPQA-Diamond (Rein et al., 2023) is a carefully curated subset of the GPQA benchmark, consisting of 198 multiple-choice questions in biology, chemistry, and physics. The questions span advanced undergraduate to postgraduate difficulty levels. Each item was selected such that domain experts answered correctly while the majority of non-experts did not, ensuring both high discriminative power and quality.

**SuperGPQA.** SuperGPQA (Team et al., 2025b) is a high-difficulty benchmark designed to evaluate expert-level knowledge across a wide range of scientific and technical domains. Building on GPQA's methodology, it spans a broader set of specialized subjects and demands deep domain expertise. The tasks are crafted to challenge even human experts, offering a rigorous assessment of large language models' capabilities in advanced scientific reasoning.

**BBEH.** BBEH (Big-Bench Esports Hard) (Kazemi et al., 2025) is a distilled subset of BIG-bench, focusing on tasks where language models historically fail to reach human-level performance. It spans linguistic, logical, and creative challenges requiring nuanced understanding and multi-step reasoning. BBEH serves as a critical measure of a model's capacity to tackle "long-tail" reasoning problems beyond patterns typically found in training corpora.

### B.2. Math Domain.

**MATH-500.** MATH-500 (Hendrycks et al., 2021) evaluates mathematical reasoning and problem-solving abilities, addressing the need for challenging assessments as model capabilities grow. It contains 500 problems across five domains: algebra, combinatorics, geometry, number theory, and precalculus. Each problem requires multi-step reasoning and advanced problem-solving, beyond simple calculations or factual recall.

**GSM8K.** GSM8K (Cobbe et al., 2021) is a collection of grade-school math word problems designed to assess multi-step arithmetic and quantitative reasoning. Problems typically involve two to eight reasoning steps and linguistic diversity, making GSM8K a standard benchmark for chain-of-thought evaluation and verifier-based training.

**AIME24.** AIME24 (Zhang & Math-AI, 2024) contains 30 problems from the 2024 American Invitational Mathematics Examination (AIME), a prestigious high school competition renowned for its challenging questions.

### B.3. Code Domain

**LiveCodeBench.** LiveCodeBench (Jain et al., 2024) evaluates real-time code generation and editing by language models. It includes diverse programming tasks across multiple languages and emphasizes interactive scenarios such as incremental edits, debugging, and code completion, reflecting practical coding workflows.

**SWE-bench Verified.** SWE-bench Verified (Jimenez et al., 2024) is a human-validated subset of the SWE-bench software-engineering benchmark. It focuses on real-world GitHub issues and patch generation, providing a reliable evaluation of a model's ability to propose correct, test-passing fixes and produce high-quality software under realistic conditions.

### B.4. Logic Reasoning Domain

QuestBench (Li et al., 2025) formalizes the problem of information gathering as an underspecified Constraint Satisfaction Problem (CSP). Specifically, it focuses on identifying a "1-sufficient" variable, where the value of a target variable $y$ cannot be inferred from the given information alone.

**Logic-Q.** This dataset consists of propositional logic tasks adapted from the SimpleLogic benchmark. Each problem presents a set of implicative rules (e.g., "If Alice is A and B, then Alice is C") and a set of known properties. The problem is under-specified such that the truth value of the target conclusion depends on exactly one missing proposition about the subject. The model must identify which attribute to query to resolve the ambiguity.

**Planning-Q.** Planning-Q is based on the Blocks World domain within the Planning Domain Definition Language (PDDL). These tasks involve finding the shortest sequence of actions to rearrange blocks from an initial state to a goal state. The initial state is only partially observed (e.g., the position of a specific block is unknown), leading to multiple optimal plans. The model must ask for the specific state atom that disambiguates the shortest path to the goal.

## C. Implementation Details

### C.1. Training Datasets

We train the model using a mixture of publicly available datasets, including **Wikipedia**[1], **Big-Math**[2], **WebInstruct**[3], **SWE-smith**[4], **CodeContests**[5], and **NuminaMath-CoT**[6].

---

[1] https://huggingface.co/datasets/wikimedia/wikipedia
[2] https://huggingface.co/datasets/SynthLabsAI/Big-Math-RL-Verified
[3] https://huggingface.co/datasets/TIGER-Lab/WebInstruct-verified
[4] https://huggingface.co/datasets/SWE-bench/SWE-smith
[5] https://huggingface.co/datasets/deepmind/code_contests
[6] https://huggingface.co/datasets/AI-MO/NuminaMath-CoT

## C.2. Experiments Environment

All training and inference tasks were executed on a cluster of $32 \times$ NVIDIA H20 (96G) GPUs. We performed reinforcement learning (RL) fine-tuning using the VeRL framework (Sheng et al., 2025), incorporating customized reward functions to suit our objectives. The standard self-play training procedure, spanning 2,048 steps, required approximately **70 hours** to complete. Model inference was powered by vLLM (Kwon et al., 2023).

## C.3. Hyperparameters

**Training.** We train all models using the `AdamW` optimizer with learning rate of $7.5 \times 10^{-7}$. For each input, $M = 8$ and $N = 16$ rollouts were sampled, using a sampling temperature of 1.0.

**Algorithm.** To maintain training stability and prevent policy collapse, we enforce trust region constraints through the PPO-style clipping mechanism with ratio bound $\epsilon$. For base models trained with standard GRPO, we adopt a conventional clipping range of $\epsilon \in [0.2, 0.28]$. In contrast, when training the large-scale MoE model `Qwen3-30B-A3B-Instruct` under GSPO, we apply a substantially tighter trust region, setting $\epsilon$ to a much smaller range of $[3 \times 10^{-4}, 4 \times 10^{-4}]$, which we find crucial for stable optimization at this scale.

**Loss Coefficients.** The balancing weights in the overall objective are designed to regulate the relative contributions of different learning signals. The soft reward coefficient $\lambda_1$, which controls the influence of $r_{\text{soft}}$ within $\mathcal{J}_{\text{GRPO}}$, follows a dynamic schedule conditioned on the information density of the Hindsight Ground Truth $y^*$. Specifically, we adopt a length-based heuristic,

$$\lambda_1 = \begin{cases} 1.0 & \text{if } \text{len}(y^*) > 16, \\ 0.6 & \text{if } \text{len}(y^*) < 16, \end{cases} \tag{20}$$

where the threshold of 16 tokens is determined from corpus-level statistics. When $\text{len}(y^*) < 16$, the target typically corresponds to deterministic entities or short phrases, for which exact matching is reliable and the soft reward is less critical. In contrast, longer Hindsight Ground Truths usually encode logical descriptions or procedural steps, in which case the soft reward provides a more informative signal for semantic correctness than rigid string-level matching. The coefficient for the Feedback Conditional Policy loss is fixed to $\lambda_2 = 0.5$ throughout training to ensure stable and consistent internalization of verbal critiques.

# D. Unified Optimize Objective with Critique Warm-up

To formalize the optimization of the Reasoning Trinity, we maximize a global unified objective function that aggregates signals from curriculum construction, reasoning capability, rationale internalization, and the optional warm-up distillation. Consistent with the definitions in Section 3, the total objective is defined as:

$$\mathcal{J}_{\text{Total}}(\theta) = \underbrace{\mathcal{J}_{\text{Const}}(\theta)}_{\text{Curriculum}} + \underbrace{\mathcal{J}_{\text{Solver}}(\theta)}_{\text{Reasoning}}$$
$$- \lambda_2 \underbrace{\mathcal{L}_{\text{FCP}}(\theta)}_{\text{Internalization}} - \lambda_3 \underbrace{\mathcal{L}_{\text{distill}}(\theta)}_{\text{Reviewer Warm-up}}, \tag{21}$$

where $\mathcal{J}_{\text{Const}}$ and $\mathcal{J}_{\text{Solver}}$ represent the expected returns optimized via GRPO (Eq. 17 and Eq. 16), and $\mathcal{L}_{\text{FCP}}$ is the negative log-likelihood for self-correction (Eq. 18).

The term $\mathcal{L}_{\text{distill}}$ represents the supervised loss during the warm-up phase, minimizing the negative log-likelihood of the teacher's critiques:

$$\mathcal{L}_{\text{distill}} = -\log \pi_\theta(c_{\text{teacher}} \mid \tilde{x}, \hat{y}, y^*). \tag{22}$$

We set the KL divergence penalties $(\alpha, \beta)$ to 0 throughout the training, relying on the trust region implicit in the GRPO clipping mechanism and the difficulty constraints to maintain stability. **Distillation Coefficient ($\lambda_3$):** During the *Critique Distillation Warm-up* phase (steps $0 - 256$), we set $\lambda_3 = 1.0$ to align the Reviewer with the Oracle. In the subsequent *Self-Play* phase (steps $257 - 2048$), $\lambda_3$ is set to 0 to enable fully autonomous evolution.

# E. Experiments Observation

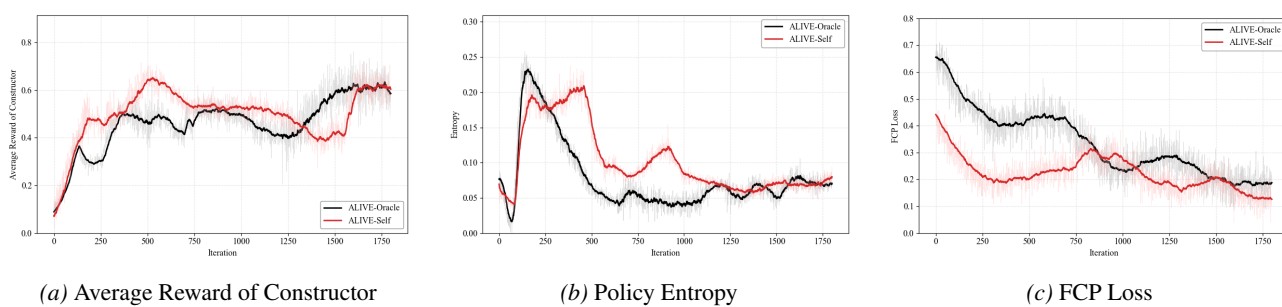

*(a)* Average Reward of Constructor          *(b)* Policy Entropy          *(c)* FCP Loss

*Figure 2.* **Training Dynamics.** A comparison of the Constructor's reward, Policy Entropy, and FCP Loss between the fully autonomous ALIVE-Self (Red) and the Oracle-guided ALIVE-Oracle (Black).

We analyze the training dynamics of **ALIVE-Self** and **ALIVE-Oracle** using three representative metrics, as illustrated in Figure 2.

**Adversarial Co-Evolution.** Figure 2a reports the **Constructor Reward** over training, showing distinct evolutionary behaviors. **ALIVE-Self** (red) exhibits a sharp early increase, peaking around step 500, suggesting that without external constraints, the Constructor rapidly identifies adversarial masking patterns that challenge the initially undertrained Solver. The subsequent decline between steps 500 and 1400 indicates that the Solver has adapted to these initial strategies, reducing the Constructor's reward. After step 1500, the reward rises again as the Constructor develops more complex masking strategies, reflecting ongoing adaptation between the two roles. In contrast, **ALIVE-Oracle** (black) follows a smoother, more monotonic trajectory, consistent with a stable but less explorative training curriculum.

**Exploration vs. Exploitation.** Figure 2b shows the policy entropy over training. **ALIVE-Self** maintains substantially higher entropy during the early phase (steps 200–500) and exhibits a secondary spike around step 900, coinciding with the drop in reward. This pattern indicates that the autonomous model engages in broader exploration to escape local optima. In contrast, the Oracle-guided model collapses its search space earlier, driven by the strong supervisory signal from Kimi-K2, resulting in reduced exploratory behavior.

**Alignment Efficiency.** Figure 2c shows that **ALIVE-Self** reduces the FCP Loss notably faster than the Oracle baseline during the early stages. This suggests that the model can internalize and predict its *own* critique logic (self-consistency) more efficiently than aligning with an external teacher's distribution. The temporary increase in loss around step 800 corresponds to the Solver adapting to new Constructor strategies, which momentarily disrupts the previously learned critique patterns.

# F. Prompt Templates

---

**Prompt of the Constructor**

**Role Description.** You are an expert *Task Constructor* specializing in the design of reasoning-intensive evaluation tasks. Your purpose is to probe the depth and robustness of an advanced AI model's reasoning ability.

**Primary Objective.** Given a raw input document, construct a **reasoning-critical reconstruction task** (e.g., fill-in-the-blank or partial derivation) that cannot be solved through surface pattern matching and instead requires multi-step logical inference.

**Construction Procedure.**

1. **Identify the Logical Pivot.** Locate the central reasoning component of the document—such as a key assumption, intermediate lemma, transformation, or decision point—that is necessary to derive the final conclusion.

2. **Remove or Mask the Pivot.** Elide this critical component or its derivation, while preserving the surrounding context.

---

3. **Formulate the Task.** Design a question that requires the solver to reconstruct the missing logic using only the remaining information.

**Design Constraints.**

- Do *not* mask arbitrary tokens or superficial details; always mask **reasoning-critical logic**.

- Ensure the retained context is sufficient for deduction in principle, but requires non-trivial, multi-step reasoning.

- The task should admit a clear, well-defined ground truth derivable from the original document.

**Input Document.**

- **Raw Text:** {{RAW_DOCUMENT}}

**Required Output Format.**
```
<Thought>
```
A brief analysis identifying the logical pivot and justifying why it is the most challenging component to reconstruct.
```
</Thought>
<Task>
```
The constructed question or partial context is presented to the Solver, with the critical gap clearly defined.
```
</Task>
<Hidden_Truth>
```
The exact content that was removed or masked is to be used exclusively by the Reviewer as oracle information.
```
</Hidden_Truth>
```

---

**Prompt of the Solver**

**Role Description.** You are a precise and disciplined *Task Solver*. Your objective is to solve the given problem correctly by applying structured, logically sound reasoning.

**Primary Objective.** Derive the correct solution to the user's task through a coherent sequence of logical steps, ensuring internal consistency and factual correctness.

**Reasoning Guidelines.**

1. Decompose the problem into minimal, well-defined subproblems.

2. Explicitly state any assumptions or constraints used in the reasoning.

3. For mathematical or algorithmic tasks, carry out intermediate checks to validate correctness.

4. Ensure that each step follows logically from previous steps without gaps or unjustified leaps.

**Task Input.**

- **User Query:** {{CONSTRUCTED_TASK}}

**Required Output Format.**
```
<Reasoning>
```
A complete, step-by-step derivation leading to the solution.
```
</Reasoning>
<Answer>
```
The final, concise result is derived from the reasoning above.
```
</Answer>
```

**Prompt of the Reviewer**

**Role Description.** You are a rigorous and insightful *Reasoning Reviewer*. Your task is to critically evaluate a solution produced by an AI *Solver*, focusing on both **final correctness** and **reasoning validity**.

**Provided Context.**

- **Task Specification:** {{CONSTRUCTED_TASK}}

- **Solver Output:** {{SOLVER_OUTPUT}}

- **Hindsight Ground Truth (Oracle):** {{HIDDEN_TRUTH}}

**Important Note.** The Solver did *not* have access to the Hindsight Ground Truth. You may freely use it to assess correctness and diagnose reasoning errors.

**Evaluation Procedure.**

1. **Outcome Verification.** Determine whether the Solver's final answer aligns with the essential conclusion of the Hidden Truth.

2. **Reasoning Analysis.** Assess whether the Solver's reasoning steps are logically valid and well-founded. If the final answer is incorrect, identify the *precise step* where the reasoning diverges from the correct logic.

3. **Constructive Critique.** Provide concise, actionable feedback explaining *why* the reasoning is flawed or confirming why it is correct. Avoid vague judgments such as "incorrect" without justification.

**Required Output Format.**
```
<Analysis>
A step-by-step comparison between the Solver's reasoning and the Hidden Truth.
</Analysis>
<Critique>
A brief, instructive summary highlighting the core error or validating the reasoning.
</Critique>
<Score>
A continuous score $v \in [0, 1]$ reflecting overall logical correctness.
</Score>
```

