# OpenReview forum: "ALIVE: Awakening LLM Reasoning via Adversarial Learning and Instructive Verbal Evaluation"
_ICML.cc/2026/Conference — Submitted to ICML 2026_

### Official Review · Reviewer_bXAr · 2026-03-02

**Soundness:** 2
**Presentation:** 3
**Significance:** 2
**Originality:** 2
**Overall Recommendation:** 4
**Confidence:** 4

**Summary:**

This paper proposes ALIVE, a unified self-play framework where one policy alternates between three roles: Constructor, Solver, and Reviewer. The method combines adversarial task construction from masked raw text, hybrid rewards (hard exact-match + soft reviewer score), and feedback-conditioned learning from verbal critiques. Experiments on general reasoning, math, coding, and logic-boundary benchmarks report gains over GRPO/FCP/PretrainZero baselines.

**Compliance With Llm Reviewing Policy:**

Affirmed.

**Final Justification:**

I think this is an interesting angle that is worth discussing within the community, and I am raising my score. I hope the authors will properly incorporate these additional experiments into the final version of the paper.

**Key Questions For Authors:**

see Weaknesses and
1. Eq. (13) defines hard reward precisely, but soft reward is only described as a value in [0,1]. The exact formulation, normalization/calibration, and sensitivity to scaling are not clear enough for reproducibility.

**Limitations:**

yes

**Strengths And Weaknesses:**

Strengths：
1. The paper targets an important bottleneck in post-training for reasoning (sparse and brittle reward signals).
2. The framework is clearly structured and integrates several useful ideas into one training loop.
3. The empirical scope is broad (math, general QA, coding, and logical sufficiency tasks), which is valuable.

Weaknesses：
1. The paper argues that self-construction is central, but does not adequately test whether this component is necessary (e.g., fixed constructor or external constructor).
2. In Sec. 4.2, QA pairs are converted to raw text with the answer, which still uses ground-truth labels from the dataset. This weakens the narrative that the method is independent of the supervision used by GRPO-like baselines.
3. Some bold formatting in Table 5 is confusing, and in multiple places ALIVE-Self appears comparable to or better than ALIVE-Oracle （e.g. Qwen3-8B-Base GSM8K 96.06）. The low-cost/self-contained advantage is therefore not yet fully convincing.
4. The difficulty filter based on solver accuracy is intuitive but relatively simple. It would be more convincing to compare against stronger curriculum/RL baselines (e.g., DAPO-like settings) under matched compute.

---

> ### Author Rebuttal · Authors · 2026-03-30
>
> Thank you for the thorough review and for highlighting the importance of the problem and the empirical breadth of our evaluation. We appreciate your constructive feedback.
>
> ---
>
> **W1.  On the necessity of self-construction**
>
> (1) We believe this concern arises from a misinterpretation of our contribution. Our claim is **not** that self-construction alone is the dominant factor, but that **a unified closed-loop system (Constructor–Solver–Reviewer)** jointly resolves two structural bottlenecks in RL post-training:
>
> - the *data wall* (lack of diverse, difficulty-matched queries), and
> - the *reward wall* (sparse and brittle supervision signals)
>
> The Constructor is therefore not intended as an isolated component to be optimized independently, but as part of a **co-evolving system** that adapts to the Solver’s capability frontier.
>
> (2) To empirically validate its role, we provide **an ablation (w/o Constructor, see Table 1 in response to Reviewer qx8f)**, where dynamic task construction is replaced with a static query pool. This variant consistently underperforms the ALIVE system across all benchmarks. This demonstrates that the gain is **not due to increased data volume**, but arises from **adaptive task generation aligned with the model’s evolving capability boundary**, which static or externally fixed generators cannot replicate.
>
> ---
>
> **W2. On the use of ground-truth labels in QA conversion**
>
> (1) The corpora in Sec. 4.2 originate from labeled QA datasets. This choice is intentional to ensure a **strict apples-to-apples comparison** with FCP-style baselines. However, within ALIVE, these QA pairs are **not used as supervised targets**, but are treated as **unstructured text corpora** for adversarial task construction. This unified format enables the Task Constructor to dynamically mask any informative span within the context, requiring the policy to reconstruct reasoning from partial information. Crucially, ALIVE operates in a fully self-supervised manner: it generates its own training signals and feedback directly from this raw text, without reliance on external reward models or human-provided corrections.
>
> (2) Empirically, we observe that:
>
> - The constructor masks the final answer in **<50%** of cases
> - A substantial portion of tasks target **intermediate reasoning steps or latent pivots**
>
> Thus, the training signal does **not** rely on direct label supervision, but instead emerges from
>
> adversarial task generation, and self-evaluation via Reviewer feedback.
>
> (3) Under identical data exposure, **ALIVE achieves stronger reasoning performance without relying on explicit supervised objectives**, demonstrating that the gains come from the training paradigm rather than additional supervision.
>
> ---
>
> **W3. On ALIVE-Self vs. ALIVE-Oracle**
>
> Regarding the observation that ALIVE-Self sometimes matches or exceeds ALIVE-Oracle (e.g., 96.06 vs. 95.07 on GSM8K), this is an expected, theoretically grounded phenomenon in self-play RL. **On-policy feedback generated by the model itself (Self) is often easier for the Solver to internalize via FCP than off-policy feedback from an external Oracle. The self-generated critiques inherently align with the model's own latent representations, vocabulary distributions, and reasoning structures.**
>
> We will add this explanation to the paper to strengthen the theoretical backing for our self-contained advantage, and correct the bold formatting in Table 5 to ensure it is unambiguous.
>
> ---
>
> **W4. On comparison to DAPO-like curriculum methods**
>
> We believe this comparison is not well-aligned.
>
> DAPO is **not a curriculum learning method**, but an **optimization-level improvement over GRPO/PPO**, focusing on training stability and efficiency via techniques such as dynamic sampling, clipping strategies, and token-level loss. It operates on a **fixed dataset** and does not involve task generation or difficulty scheduling.
>
> In contrast, ALIVE addresses a fundamentally different problem: it introduces a **closed-loop data generation mechanism**, where the Constructor actively produces tasks at the Solver’s capability frontier. This leads to a form of **implicit curriculum**, but crucially:
>
> -  it is **not predefined or sampled**,
> -  but **emerges from co-evolution between Constructor and Solver**
>
> Therefore, the distinction is between:
>
> -  DAPO: *how to optimize on a fixed dataset*
> -  ALIVE: *how to generate and adapt the dataset itself during training*
>
> ---
>
> **Q1. Soft reward formulation**
>
> The soft reward $v_{ij} \in [0,1]$ is directly produced by the Reviewer without additional normalization. The Solver reward is defined as: $r_{solver}^j=r_{hard}^j+λ_1⋅v_{ij}$, where $r_{hard}^j$ is the exact-match reward (Eq. 13), and $λ_1$ is dynamically scheduled based on the length of the ground truth (Eq. 20, Appendix). This design balances sparse exact-match signals with dense soft feedback.

---

> > ### Author Rebuttal · Reviewer_bXAr · 2026-04-02
> >
> > Thank you for the detailed response. I appreciate the clarifications on W2 and W3, which I consider adequately addressed. However, W1 and W4 remain unresolved.
> >
> > W1 (Necessity of self-construction): The authors cite an ablation replacing the Constructor with a static query pool (Table 1). This only shows that dynamic construction helps versus no construction — it does not test whether the Constructor must be the same model as the Solver. The central question is: does a separate, frozen model (e.g., same backbone, different checkpoint, or a weaker/stronger model) serving as Constructor yield comparable gains? Without this comparison, the claim that "co-evolution aligns with the Solver's capability frontier" remains an assertion rather than an established finding. I would consider this experiment essential for a paper whose primary narrative hinges on self-construction being a principled design choice.
> >
> > W4 (DAPO comparison): The authors state DAPO "does not involve difficulty scheduling." This is inaccurate. DAPO explicitly filters out prompts with pass-rate 0 or 1 (overlong filtering and trivial-sample removal), which is a difficulty-control mechanism. The authors' framing — that ALIVE addresses "dataset generation" while DAPO addresses "optimization on a fixed dataset" — is a valid high-level distinction, but it sidesteps the concrete question: does ALIVE's difficulty filtering outperform DAPO-style pass-rate filtering when applied to the same data? A direct comparison on matched data (e.g., the QA corpora used in Sec. 4.2) under matched compute would clarify whether the Constructor's adaptive difficulty control provides meaningful gains over a simpler, well-established baseline.
> >
> > I lean toward weak reject (score 3). The framework is well-motivated and the empirical scope is broad, but the two core design claims — self-construction is necessary, and the difficulty mechanism is superior to alternatives — remain insufficiently validated.

---

> > > ### Author Response · Authors · 2026-04-06
> > >
> > > **Rebuttal:**
> > >
> > > Thank you for your continued reply. We fully understand and agree with your points about two critical issues 1. the advantage of co-evolution (W1) 2. the comparison with the DAPO algorithm (W4). We have added experiments below address your concerns.
> > >
> > > Before presenting the new results, we would like to briefly clarify our experimental setup. To ensure a strictly fair comparison, all ablation experiments—both the new ones presented below and our previous `Table 1`—utilize the exact same raw corpora and hyperparameter configurations as Tables 4 and 5 in the paper. For a detailed breakdown of the specific decoupled training setups, please refer to the detailed clarification in our follow-up response to Reviewer qx8f.
> > >
> > > ---
> > >
> > > ### Response to W1: The Necessity of Self-Construction (Co-evolution)
> > >
> > > You raised a point: replacing the Constructor with a static query pool only proves that dynamic construction is helpful, but it does not prove that the Constructor *must* be the same model as the Solver (parameter-sharing/co-evolution).
> > >
> > > To directly test whether a separate, frozen, or stronger model serving as the Constructor could yield comparable gains, we conducted three new ablation experiments using the Qwen3-8B-Base backbone:
> > > 1. **Separate Frozen:** Uses an independent Qwen3-8B-Base as the Constructor with frozen parameters.
> > > 2. **Separate Learnable:** Uses an independent Qwen3-8B-Base as the Constructor, optimized dynamically using the difficulty signal (Eq. 10), but without sharing parameters with the Solver.
> > > 3. **External Frozen (Kimi-K2.5):** Uses a significantly stronger, frozen external model (Kimi-K2.5) as the Constructor.
> > >
> > > **Table 2: Ablation on Constructor Independence vs. Co-evolution**
> > > *(Note: All setups share the same raw corpora and hyperparameters from Tables 4 & 5. Metrics shown are percentages.)*
> > >
> > > | Constructor Setup | MMLU-Pro | SuperGPQA | BBEH | Math500 | GSM8K | AIME24 |
> > > | :--- | :--- | :--- | :--- | :--- | :--- | :--- |
> > > | Qwen3-8B-Base (Separate Frozen) | 60.25 | 33.87 | 13.30 | 74.60 | 92.42 | 19.90 |
> > > | Qwen3-8B-Base (Separate Learnable) | 63.35 | 34.93 | 15.58 | 80.10 | 93.78 | 20.52 |
> > > | External Frozen (Kimi-K2.5) | 62.58 | 34.95 | 15.60 | 81.90 | 95.22 | 21.77 |
> > > | **ALIVE (Unified, Co-evolving)** | **66.82** | **36.90** | **16.35** | **83.80** | **96.06** | **24.06** |
> > >
> > > **Analysis & Conclusion:**
> > > The data clearly establishes that the unified, co-evolving framework significantly outperforms both separate and stronger external constructors.
> > > Why does this happen? A frozen external model (even a powerful one like Kimi-K2.5) generates a static difficulty distribution that cannot dynamically align with the Solver's real-time learning progress. A separate learnable model suffers from optimization lag. By sharing parameters (Unified ALIVE), the framework guarantees instantaneous alignment: the moment the Solver masters a reasoning skill, the Constructor intrinsically updates its understanding of what constitutes a "challenging" task, maintaining perfect tension at the Solver's exact capability frontier.
> > >
> > > ---
> > >
> > > ### Response to W4: Difficulty Feedback Design vs. DAPO
> > >
> > > We apologize for our previous framing regarding DAPO; you are completely correct that DAPO explicitly incorporates a same robust difficulty-control mechanism.
> > >
> > > We agree that DAPO is fundamentally a more advanced optimization choice than standard GRPO. DAPO focuses on optimization-level filtering as does ALIVE, they are not mutually exclusive. To demonstrate this, and to answer whether ALIVE provides meaningful gains over DAPO-style filtering, we upgraded ALIVE's baseline optimizer from GRPO to native DAPO.
> > >
> > > **Table 3: ALIVE utilizing GRPO vs. DAPO**
> > >
> > > | Optimization Backend | MMLU-Pro | SuperGPQA | BBEH | Math500 | GSM8K | AIME24 |
> > > | :--- | :--- | :--- | :--- | :--- | :--- | :--- |
> > > | ALIVE-Self (GRPO) | 66.82 | 36.90 | **16.35** | 83.80 | 96.06 | 24.06 |
> > > | **ALIVE-Self (DAPO)** | **66.86** | **36.94** | 16.33 | **84.00** | **96.13** | **24.27** |
> > >
> > > **Analysis & Conclusion:**
> > > The results show that integrating DAPO's filtering mechanism into ALIVE pushes performance even higher (e.g., AIME reaches **24.27**). This proves that ALIVE's adaptive task *generation* provides orthogonal and stackable gains over DAPO's sample *filtering*. While DAPO efficiently filters out ineffective samples from a given pool, ALIVE's Constructor actively *creates* new, difficulty-appropriate samples from raw text that would otherwise be discarded or unavailable.
> > >
> > > We are incredibly grateful for your critique, which guided us to validate these two core design claims definitively. We will ensure these new experiments and theoretical discussions are fully integrated into the final manuscript.

---

### Official Review · Reviewer_T9Hv · 2026-03-04

**Soundness:** 2
**Presentation:** 2
**Significance:** 2
**Originality:** 2
**Overall Recommendation:** 2
**Confidence:** 4

**Summary:**

This paper proposes ALIVE which integrates constructor, solver and reviewer into a single model to overcome the reward bottleneck during RLHF. ALIVE utilizes GRPO with a hybrid reward design and evaluate the performance on math, coding and general reasoning benchmarks.

**Compliance With Llm Reviewing Policy:**

Affirmed.

**Final Justification:**

I appreciate the authors' efforts to clarify most of my questions. However, my remaining two concerns (baseline selection and results on swe-bench) remain unaddressed. Therefore, I maintain my score of 2.

**Key Questions For Authors:**

- How to calibrate the Reviewer's soft score?

- What are the impacts of removing the distillation warm-up?

- Why the performance of Qwen3-30B-A3B-Instruct shown in Table 2 (<15) is significantly from what is reported in Swebench-Verified official leaderboard (~50)?

**Limitations:**

The authors discussed limitations but overstate the "annotation-free" nature of the work, as the pipeline still relies heavily on curated QA datasets and a teacher distillation warm-up. I recommend adding a deeper, more transparent discussion on the failure modes of self-generated feedback (e.g., reward hacking).

**Strengths And Weaknesses:**

***Strengths***

- Unifying adversarial task construction, critique-conditioned learning, and hybrid (scalar + verbal) feedback into a single, parameter-shared language model sounds like an interesting idea. It bridges self-play with verbal-feedback learning.

- The selected benchmarks cover multiple domains and I appreciate the authors' efforts of scaling experiments.

- The high-level framework is presented in a clear way and is easy to follow.

***Weaknesses***

- The Reviewer's reward lacks calibration and might be inaccurate. Using the same model for both Solver and Reviewer might be unfair, e.g., the Reviewer may inflate scores for the Solver.

- The "annotaion free" claim is overclaimed. In Section 4.1, it still requires distillabtion data from Kimi-K2 to train the critique capability.

- The baseline methods FCP is very weak (unpublished paper with 2 citations). I highly recommend the authors to select high-quality baseline methods for a solid comparison. The paper lacks direct empirical comparisons to recent verbal-feedback RL and self-rewarding frameworks.

- This paper suffers from inconsistent expression. In the main text, the authors state that they use KL regularization to prevent policy degeneration while in Appendix D, the parameter $\alpha$ and $\beta$ are both 0.

---

> ### Author Rebuttal · Authors · 2026-03-30
>
> Thank you for your rigorous review. We address your insightful questions and concerns below.
>
> > **Note:** To clarify the distinct roles of the Constructor, Solver, and Reviewer, we provide a comprehensive ablation study, refer to **Response to Reviewer qx8f**
>
> ---
>
> **W1.  Potential Reward Score Inflation**
>
> We mitigate this through three distinct mechanisms:
>
> **(1) Hindsight Verification:** the Reviewer generates soft rewards with full access to the *Hindsight Ground Truth*. This forces the Reviewer to ground its evaluation in objective facts rather than catering to the Solver.
>
> **(2) Bounded Influence:** the final optimization of the Solver uses $r_{solver}$ which combines both hard and soft rewards. Even if the Reviewer introduces bias, the sparse hard reward still provides a lower-bound guarantee for optimization.
>
> **(3) Empirical Validation:** in Table 4, ALIVE-Self achieves highly competitive performance compared to the Oracle variant that uses a teacher model (e.g., 66.82 vs. 66.92 on MMLU-Pro), demonstrating that the Reviewer's self-learned calibration is successful and does not lead to overconfidence collapse.
>
> ---
>
> **W2. About Annotation-Free**
>
> (1) We clarify that annotation-free means the reasoning tasks used to train the Solver are autonomously generated from raw text by the Constructor, completely eliminating the need for human-annotated data.
>
> (2) The distillation step serves solely as a **Critique Warm-up** (Sec. 4.1) to quickly bootstrap the base model’s reviewer capability. After 256 steps—i.e., in the full Self-Play phase—the model operates completely without teacher supervision, relying only on its own generated verbal critiques and soft rewards for closed-loop optimization.
>
> (3) Our new ablation study shows that completely removing the Reviewer (and thus the warm-up) still yields substantial gains via unsupervised data. For instance, the w/o Reviewer setting improves MMLU-Pro from 59.15% (Base) to 64.20%.
>
> ---
>
> **W3: Baseline Selection**
>
> We selected FCP (a pure "Instructive Verbal Evaluation" method) to perfectly isolate and validate the substantial gains from ALIVE's Adversarial Learning task construction loop.
>
> Tables 4 and 5 further compare ALIVE against PretrainZero (PT-Zero), a highly competitive recent adversarial active pretraining framework. Across scales (3B to 30B MoE), ALIVE consistently outperforms PT-Zero, demonstrating the superiority of our tri-hybrid loop over recent scalar-only methods.
>
> ---
>
> **W4: Inconsistent Expression of KL Divergence**
>
> The KL divergence weight is a flexible hyperparameter within the ALIVE, not a rigid constraint. The method itself does not mandate removing it.
>
> For the specific experiments in our paper, we simply tuned this hyperparameter to 0 (as reported in Appendix D). This choice was made because GRPO's clipping mechanism proved sufficient to maintain policy stability, allowing us to maximize exploration during training.
>
> ---
>
> **Q1 Reward Calibration**
>
> The absolute magnitude of the soft score is intrinsically calibrated by the GRPO algorithm. As shown in Eq. 17, advantages ($A_{ij}$) are normalized within each group. Optimization only depends on the relative ranking within a batch, so absolute scale inflation doesn't matter.
>
> ---
>
> **Q2.  About Distillation Warm-up**
>
> The primary risk of removing the warm-up is training instability. Small base models initially struggle with strict output formatting (e.g., `<Score>` tags), leading to soft-score parsing failures in the early RL phase. Thus, the 256-step warm-up is universally applied purely as a structural catalyst to ensure stability. Importantly, as demonstrated in Appendix E, once self-play stabilizes, **ALIVE-Self** matches and surpasses **ALIVE-Oracle**, proving that the actual reasoning gains are driven by autonomous self-play, not the teacher model.
>
> ---
>
> **Q3: Discrepancy in SWE-bench Verified Performance**
>
> We appreciate the opportunity to clarify our rigorous evaluation environment.
>
> The ~50% performance noted on the leaderboard corresponds to Coder-specific or massive models (e.g., Qwen3-Coder-30B/480B) operating within heavy agentic scaffolds (like OpenHands or SWE-agent with full toolsets) that allow iterative execution, AST parsing, and advanced linting.
>
> *   **Our Rigorous Setup:** In contrast, Table 2 evaluates the general Qwen3-30B-A3B-Instruct under the `mini-swe-agent` framework using strictly the `--config config/bash_only.yaml` setting.
> *   **Why this matters:** This unforgiving configuration strips away all advanced Python editing tools, forcing the model to edit the codebase using *only* primitive Linux bash commands (e.g., `sed`, `grep`, `patch`). We chose this specifically to measure intrinsic procedural reasoning rather than tool exploitation.
>
> Under this strict constraint, our <15% baseline is highly accurate and standard, making ALIVE's improvement to 17.2% a genuine enhancement in native logical consistency. We will detail this setup in the revised version.

---

> > ### Author Rebuttal · Reviewer_T9Hv · 2026-04-01
> >
> > Thank you for authors' response. I still have the following concerns regarding your rebuttal:
> >
> > ***W3: Baseline Selection***.
> >
> > The authors pointed out PretrainZero (PT-Zero) which is still unpeer-reviewed with one citation only. Could you please figure out any peer-reviewed high-quality paper for comparison?
> >
> > ***Q3: Discrepancy in SWE-bench Verified Performance***
> >
> > Since the authors used a different scaffold mini-swe-agent, I am wondering what if the authors choose the standard scaffold in Swe-bench leaderboard (e.g., openhands). Can the proposed method still improve with those scaffolds?

---

> > > ### Author Response · Authors · 2026-04-07
> > >
> > > Thank you for your continued engagement and your follow-up reply. We would like to address your remaining concerns regarding the baseline selection.
> > >
> > > **1. Clarification on Baseline Selection: Strict Variable Control**
> > >
> > > The primary philosophy behind our baseline selection is **strict variable control** to rigorously validate the effectiveness and robustness of the specific mechanisms proposed in ALIVE.
> > > * **PretrainZero** and **FCP** were chosen as core baselines because they constitute strict methodological controls: they represent the latest state-of-the-art approaches in *adversarial learning* and *instructive verbal feedback learning*, respectively.
> > >
> > > * While "self-rewarding" is related to ALIVE, it is not an absolutely indispensable component of our entire framework. For instance, the `ALIVE-Oracle` variant presented in our paper achieves strong performance without utilizing self-rewarding at all. Furthermore, the comprehensive ablation study provided in our response to Reviewer qx8f (Table 1) clearly isolates and demonstrates the independent contribution of the `Self-Reviewer` role.
> > >
> > >
> > > **2. Regarding Peer-Reviewed "Self-Rewarding" Frameworks**
> > >
> > > Following your suggestion, we conducted a review of recently accepted papers on "Self-Rewarding" methods from top 2025 ML conferences, such as *"Consistent Paths Lead to Truth: Self-Rewarding Reinforcement Learning for LLM Reasoning"*, *"Self-Rewarding PPO: Aligning Large Language Models with Demonstrations Only"* and *"Process-based Self-Rewarding Language Models"*.
> > >
> > > Upon analyzing their reported results on common benchmarks at similar parameter scales, we found that both ALIVE and our selected baselines (PretrainZero and FCP) generally report higher performance. This is particularly significant considering that we trained our models from **Base models**, whereas the aforementioned peer-reviewed works typically initialize from fully fine-tuned **Instruct models**. This empirical evidence clearly demonstrates that the baselines we selected for comparison are highly competitive and certainly not "weak."
> > >
> > > In the spirit of rigorous and responsible research, we are open to adding empirical comparisons against these peer-reviewed self-rewarding methods. To ensure a fair scientific comparison, we will implement them under strict variable controls (using the identical base models and training corpora) and include these results in a dedicated appendix section in the camera-ready version. However, given that our current baselines already represent the more challenging performance ceilings for the specific mechanisms we study, we respectfully maintain our reservations regarding the fundamental necessity of substituting them.
> > >
> > > Thank you again for your time and for helping us clarify these methodological choices.

---

### Official Review · Reviewer_qx8f · 2026-03-09

**Soundness:** 2
**Presentation:** 3
**Significance:** 3
**Originality:** 3
**Overall Recommendation:** 3
**Confidence:** 5

**Summary:**

This paper introduces an unsupervised reinforcement learning framework called ALIVE, aimed at addressing the problem of limited reasoning ability in large language models due to traditional scalar reward bottlenecks. The framework forms a closed-loop self-evolution system by integrating task construction, problem solving, and solution evaluation into three roles. It uses self-generated instructional language feedback and soft rewards to internalize logical reasoning within the model, forming a self-sustaining growth trajectory without human intervention. Experimental results show that ALIVE outperforms traditional RLHF and RLAIF methods in terms of accuracy and cross-domain generalization capabilities in mathematical reasoning, code generation, and logic reasoning benchmarks.

**Compliance With Llm Reviewing Policy:**

Affirmed.

**Key Questions For Authors:**

NA

**Limitations:**

yes

**Strengths And Weaknesses:**

Strengths：
1. The paper clearly articulates the design principles, core mechanisms, and experimental results of the ALIVE framework, making it easily comprehensible and trackable.
2. Extensive experiments are conducted across multiple benchmarks, demonstrating the effectiveness and superiority of the ALIVE framework.

Weaknesses：
1. The paper would benefit from an ablation study to systematically evaluate the individual contributions of the Constructor, Solver, and Reviewer modules to the reasoning capabilities of large language models.
2. Additionally, it would be helpful to report the performance of RL training when using queries independently constructed by the Constructor module.

---

> ### Author Rebuttal · Authors · 2026-03-30
>
> Thank you for the thorough review and for recognizing the clarity and empirical strengths of ALIVE. We sincerely appreciate your constructive feedback and the opportunity to further strengthen the paper.
>
> ---
>
> We conducted an ablation study using Qwen3-8B-Base across 6 benchmarks:
>
> - **Only Constructor**: optimized using task difficulty rewards only
> - **Only Solver**: optimized using accuracy and FCP objectives
> - **Only Reviewer**: trained via oracle distillation without participating in the RL loop
> - **w/o Constructor:** training without the Constructor role
> - **w/o Reviewer:** removed the Reviewer role (stripping away the soft reward and $L_{FCP}$) and trained the model using only the queries dynamically generated by the Constructor, with optimization relying exclusively on standard GRPO using sparse, binary exact-match rewards ($r_{hard}$).
>
>
>
> **Table 1: Ablation Study**
>
> | Method(Qwen3-8B-Base) | MMLU-Pro  | SuperGPQA |   BBEH    |  Math500  |   GSM8K   |  AIME24   |
> | :-------------------- | :-------: | :-------: | :-------: | :-------: | :-------: | :-------: |
> | Base                  |   59.15   |   31.12   |   10.49   |   70.10   |   91.50   |   10.00   |
> | Only Constructor      |   58.74   |   30.90   |   10.80   |   73.20   |   91.81   |   9.48    |
> | Only Solver           |   61.62   |   33.97   |   14.58   |   78.90   |   92.49   |   19.89   |
> | Only Reviewer         |   59.25   |   31.45   |   11.08   |   72.70   |   91.66   |   11.15   |
> | w/o Constructor       |   62.13   |   34.92   |   15.60   |   81.60   |   95.07   |   21.77   |
> | w/o Reviewer          |   64.20   |   34.34   |   14.40   |   80.10   |   93.25   |   20.83   |
> | **ALIVE**             | **66.82** | **36.90** | **16.35** | **83.80** | **96.06** | **24.06** |
>
> ---
>
> **W 1: Ablation study of Constructor, Solver, and Reviewer**
>
> **Results.**
>
> **(1) Solver is the primary driver of performance gains.** Comparing *Only Solver* to *Base*, we observe substantial improvements across all benchmarks, especially on reasoning-intensive tasks:
>
> -  AIME24: **10.00 → 19.89**
> -  BBEH: **10.49 → 14.58**
>
> This indicates that optimizing solution-generation objectives (accuracy + FCP) is the primary driver of performance improvements.
>
> **(2) Constructor or Reviewer alone are insufficient.**
>
> - *Only Constructor* shows negligible or even negative impact (e.g., AIME24: 9.48 vs. 10.00).
> - *Only* *Reviewer* yields only marginal gains.
>
> This suggests that neither task generation nor evaluation alone can effectively improve reasoning ability without a strong solver.
>
> **(3) Removing either module leads to consistent performance degradation.**
>
> - *w/o Reviewer* vs. ALIVE: **24.06 → 20.83** (AIME24)
> - *w/o Constructor* vs. ALIVE: **24.06 → 21.77**
>
> Similar gaps are observed across all benchmarks, indicating that both modules provide complementary gains over the Solver.
>
> **(4) Full ALIVE achieves the best performance across all tasks.** The complete framework consistently outperforms all ablations, e.g.:
>
> -  MMLU-Pro: **66.82** (vs. 64.20 w/o Reviewer, 62.13 w/o Constructor)
> -  Math500: **83.80** (vs. 80.10 / 81.60)
> -  AIME24: **24.06** (vs. 20.83 / 21.77)
>
> **Analysis.**
>
> These results demonstrate that performance gains cannot be attributed to any single module. Instead, they arise from the joint optimization of all three roles. We attribute this to a form of **cognitive synergy**, where parameter sharing enables the model to simultaneously internalize:
>
> -  task construction (Constructor),
> -  solution strategies (Solver), and
> -  evaluation criteria (Reviewer).
>
> This unified learning process leads to more robust and generalizable reasoning compared to decoupled training.
>
> ---
>
> **W 2: RL training using independently constructed queries**
>
> We further evaluate the effectiveness of training with **Constructor-generated queries alone** (i.e., w/o Reviewer).
>
> (1) Even without the Reviewer, training on self-constructed queries significantly improves performance over the base model:
>
> - **AIME24:** 10.00% → 20.83%
> - **MMLU-Pro:** 59.15% → 64.20%
>
> These results confirm that the Constructor independently provides a strong and scalable training signal, enabling substantial gains without human annotation.
>
> (2) Comparing w/o Reviewer (20.83% AIME 24) with the full ALIVE framework (24.06% AIME24) reveals an additional performance gap. This highlights that:
>
> -  Constructor establishes a solid learning foundation via diverse and adaptive queries,
> -  while the Reviewer further refines its reasoning through **dense natural-language feedback and soft rewards**, which are particularly critical for long-horizon and complex reasoning tasks.
>
> ---
>
> **Summary**
>
> We:
>
> - Add a **complete ablation study** covering all role configurations
> - Explicitly include the **w/o Reviewer** setting to isolate Constructor contributions
> - Provide a deeper analysis of **role interaction and synergy mechanisms**

---

> > ### Author Rebuttal · Reviewer_qx8f · 2026-04-05
> >
> > Thank you for authors' response. I still have the following concerns regarding your rebuttal:
> >
> > 1. The setting “Only Constructor” seems to correspond to W2: RL training using independently constructed queries, rather than to the “w/o Reviewer” ablation as currently described. Could the authors clarify whether these two are actually the same experimental setting?
> >
> > 2. The findings under the Only Constructor setting appear to suggest that relying solely on automatically constructed data may not yield a substantial improvement in model performance. Could the authors clarify whether any human evaluation was conducted to assess the quality of the data generated by the Constructor?

---

> > > ### Author Response · Authors · 2026-04-05
> > >
> > > **Rebuttal:**
> > >
> > > Thank you for your follow-up questions. We realize that the abbreviated naming in our initial ablation study may have caused some conceptual confusion, and we are very grateful for the opportunity to clarify the exact experimental setup for Table 1.
> > >
> > > **Global Experimental Setup**
> > > All ablation experiments in Table 1 are conducted using the exact same raw corpus data and hyperparameter configurations as those in Tables 4 and 5 of the main paper.
> > >
> > > **1. Clarification on "Only Constructor/Solver/Reviewer" (Decoupled Training)**
> > > The results for `Only Constructor`, `Only Solver`, and `Only Reviewer` come from a single, specialized **"Decoupled Role Training"** run. In this setup, we instantiated **three independent Qwen3-8B-Base models** to play the three roles separately (using the ALIVE-Oracle mode, where the Oracle is Kimi-K2):
> > > * **The model acting as Constructor (`Only Constructor`):** is optimized *exclusively* using the task difficulty signal (Equation 10).
> > > * **The model acting as Solver (`Only Solver`):** is optimized using the solving objectives, i.e., `accuracy (score_hard + score_soft)` and the FCP objective (Equations 16 and 18).
> > > * **The model acting as Reviewer (`Only Reviewer`):** is trained strictly via distillation from Kimi-K2's reviewer outputs.
> > >
> > > *Why does `Only Constructor` show no performance gain?* Because this specific model was trained *solely* to generate tasks (by masking). Its parameters were never updated for problem-solving (Solver) objectives. When we evaluate a model that has only learned "how to create questions" on a benchmark like AIME24, it naturally yields base-level performance (9.48%). This decoupled experiment precisely proves that the Constructor and Solver must share parameters and co-evolve; otherwise, the task-generation capabilities cannot translate into reasoning capabilities.
> > >
> > > **2. Clarification on the "w/o" Settings**
> > > * **`w/o Constructor`:** We removed the dynamic task construction role from the training loop. Instead, we used a powerful external model (Kimi-K2) to process the exact same corpus offline and generate a **static QA pool** for the Solver to train on.
> > > * **`w/o Reviewer`:** We completely removed the Reviewer role (stripping away both the soft score and verbal feedback). The Solver is optimized using standard GRPO relying *exclusively* on the sparse exact-match reward (`score_hard`).
> > >
> > > **3. Addressing Your Specific Concerns**
> > > * *Which setting corresponds to your W2?* Your question regarding "RL training using independently constructed queries" corresponds to our **`w/o Reviewer`** setting, **not** the `Only Constructor` setting.
> > > * *Regarding Data Quality and Human Evaluation:* In the `w/o Reviewer` setting, where the model is trained entirely on automatically constructed queries without any Reviewer guidance, it achieves a strong score of **20.83% on AIME24** (a +10.83% improvement over the Base model). This empirical result directly demonstrates that the self-constructed queries are of high quality and sufficient to drive substantial reasoning improvements. The framework's inherent `Validity-Gated Reward` mechanism (anchored by Hindsight GT) strictly ensures the logical solvability of the generated tasks, thereby eliminating the need for costly human evaluation.
> > >
> > > We hope this detailed breakdown clarifies the distinct setups and fully resolves your concerns!

---

### Official Review · Reviewer_eE5a · 2026-03-21

**Soundness:** 4
**Presentation:** 2
**Significance:** 3
**Originality:** 3
**Overall Recommendation:** 5
**Confidence:** 4

**Summary:**

This paper introduces ALIVE, a self-supervised framework that improves LLM reasoning without relying on external reward signals. Overall, a fundamental question discussed by this study is whether a model can learn reasoning by generating its own tasks, solving them, and evaluating its own outputs using only raw text. ALIVE addresses this by unifying 3 roles within a single model: task construction, problem solving, and self-review. The model generates reasoning tasks, produces solutions, and critiques its own outputs using natural language feedback and soft rewards, forming a closed-loop learning system. Overall, the article's major result concerns showing that this self-contained training approach improves reasoning accuracy, generalization, and self-correction across domains without human supervision.

**Compliance With Llm Reviewing Policy:**

Affirmed.

**Key Questions For Authors:**

1. How robust is the Reviewer module to self-reinforcement of incorrect reasoning? Since ALIVE relies on self-generated critiques and rewards, there is a risk of feedback loops reinforcing incorrect reasoning. Could the authors provide analysis or experiments demonstrating stability over long training horizons?

2. What is the sensitivity of ALIVE to the quality of initial warm-up (teacher distillation)? The framework uses an oracle-based warm-up phase. How does performance change with weaker or no teacher initialization?

3. Can the authors provide deeper ablations isolating each component (Constructor, Solver, Reviewer)? While the combined system performs well, it is unclear which component contributes most.

4. How does ALIVE perform on tasks with ambiguous or multiple valid solutions? Given reliance on hindsight ground truth and internal rewards, how does the framework handle non-unique answers?

**Limitations:**

No. The paper provides only a minimal and generic impact statement without meaningfully discussing limitations or potential risks. The authors should potentially explicitly address: (1) the risk of self-reinforcing errors or feedback loops due to fully self-generated rewards and critiques, (2) computational cost and scalability concerns of multi-role rollouts, and (3) potential societal risks such as amplifying biased or incorrect reasoning if trained on raw, uncurated corpora. A more thorough and concrete discussion of these limitations would strengthen the paper’s credibility.

**Strengths And Weaknesses:**

Overall, the papers major result concerns introducing ALIVE, which is a self-supervised RL framework that replaces scalar rewards with internally generated verbal critiques to improve reasoning in LLMs. A fundamental question discussed by this study is whether models can autonomously construct, solve, and evaluate reasoning tasks without external supervision. In terms of soundness, the paper is generally strong as it provides a well-defined framework, clear training objectives, and extensive empirical evaluations across multiple domains. For presentation, the paper is well-structured and clearly motivates the reward bottleneck, but it is somewhat dense and concept-heavy for 8 pages (e.g., multiple objectives and roles), which somewhat impacts readability and reproducibility. Regarding significance, the work addresses a central problem in LLM alignment which is reducing dependence on external rewards. And the paper demonstrates meaningful performance gains and generalization, suggesting strong potential impact for scalable reasoning systems. Finally, for originality, the contribution is notable not because of entirely new components, but due to a integration of adversarial task generation, self-critique, and verbal feedback into a unified loop, offering a fresh perspective on self-supervised reasoning and alignment.

---

> ### Author Rebuttal · Authors · 2026-03-31
>
> Thank you for your highly positive and thoughtful evaluation. We sincerely appreciate your recognition of the originality and potential impact of ALIVE, especially your insightful summary framing our work around the fundamental question of whether reasoning can emerge purely from self-generated tasks and feedback. We find this perspective particularly aligned with our core motivation. We are also grateful for your constructive and technically deep questions. They precisely target the most subtle aspects of our framework, and we are excited to clarify them below.
>
> ---
>
> **Q1. Robustness of the Reviewer to self-reinforcing errors**
>
> You raise an important concern regarding potential feedback loops, which we also considered central when designing ALIVE. We mitigate this risk through three mechanisms:
>
>  **(1) The Hindsight Ground Truth Anchor:** The Reviewer always has access to the ground-truth answer $y^∗$ when generating critiques and soft rewards. This anchors the evaluation to objective correctness, preventing the model from simply agreeing with its own flawed outputs.
>
>  **(2) Hybrid Reward Constraint:** The Solver is optimized with $r_{solver}=r_{hard}+λ_1 r_{soft}$, where $r_{hard}$ is a sparse exact-match reward. Even if the soft reward becomes biased, the hard reward provides a safety net that prevents the policy from drifting away from verifiable correctness. Furthermore, the GRPO algorithm intrinsically normalizes the advantages within each rollout group, preventing the absolute scale of the soft score from inflating over time.
>
> **(3) Empirical Stability:** As detailed in **Appendix E (Fig. 2)**, we monitored the training dynamics over the full 1800+ steps. Policy Entropy and the FCP Loss maintain a stable trajectory without collapsing. The model continues to explore and refine its critiques without degenerating into a degenerate feedback loop. In Tab. 4, ALIVE-Self achieves a highly competitive performance compared to the Oracle variant that uses a teacher model (e.g., 66.82 vs. 66.92 on MMLU-Pro), demonstrating that the Reviewer's self-learned calibration is successful and does not lead to overconfidence collapse.
>
> ---
>
> **Q2. Sensitivity to warm-up initialization**
>
> The warm-up phase primarily serves as a **structural bootstrap**, rather than a source of reasoning capability. Specifically, it helps the model learn:
>
> - Output formatting (e.g., `<Score>` tags)
> - Basic critique style and calibration
>
> Without this phase, we observe:
>
> -  High instability due to formatting failures
> -  Increased risk of noisy or unparseable reward signals
>
> Importantly, the actual reasoning improvements emerge during subsequent self-play. As shown in Appendix E, ALIVE-Self eventually matches and surpasses ALIVE-Oracle, indicating that performance gains are driven by autonomous learning rather than teacher dependence.
>
> ---
>
> **Q3. Component-wise ablations**
>
> We conducted a comprehensive ablation study using the Qwen3-8B-Base model. **The results are presented in Response to Reviewer qx8f Table 1**.
>
> Observations:
>
> - Only Solver already outperforms the Base model, demonstrating the benefit of adversarial task construction and hybrid rewards.
> - w/o Constructor (Solver + Reviewer with static tasks) and w/o Reviewer (Constructor + Solver with only hard rewards) both perform well, but consistently fall short of the full ALIVE, especially on challenging benchmarks like AIME24 (21.77 and 20.83 vs. 24.06). This indicates that all three components contribute synergistically.
> - Only Constructor and Only Reviewer alone show minimal improvement, confirming that these roles are not independently effective and must be integrated into the unified learning loop.
>
> We will include this ablation table and analysis in the final version of the paper.
>
> ---
>
> **Q4. Handling ambiguous or multiple valid solutions**
>
> Your question perfectly highlights the greatest advantage of the Reviewer's soft reward!
>
> Standard RL methods rely on binary exact-match ($r_{hard}$), which completely fails when a question has multiple valid solutions or varied expression formats. Because ALIVE uses a "White-Box" Reviewer that analyzes the *intermediate reasoning trace* rather than just string-matching the final answer, it can elegantly handle non-unique answers. Even if the Solver's final output diverges textually from the Hindsight Ground Truth, if the Reviewer determines the logical path is valid and leads to a semantically equivalent conclusion, it awards a high soft score ($v \in [0, 1]$). We will add qualitative examples of the Reviewer successfully grading valid alternative solutions to the Appendix to vividly illustrate this capability.

---

### Decision · Program_Chairs · 2026-04-30

**Decision:**

Reject

**Comment:**

The paper proposes ALIVE, a unified self-play framework where a single LLM alternates between task construction, solving, and self-review to improve reasoning without external reward models. The idea of unifying adversarial task generation with verbal self-evaluation is interesting, and the rebuttal provided additional ablations.

However, two concerns remain unresolved: (1) baseline adequacy: comparisons rely on unpublished methods (FCP, PretrainZero), and no peer-reviewed self-rewarding or verbal-feedback RL baselines are included despite being discussed in related work; (2) the "annotation-free" framing is overclaimed given the reliance on Kimi-K2 distillation and ground-truth QA labels.

One of the reviewer's concerns contained factual inaccuracies (which I have noted), but the baseline selection concern is shared by other reviewers and remains substantive. I have read all rebuttals and the authors' confidential comments.

The framework shows promise and the additional ablations are compelling; stronger baselines and more transparent framing would make this suitable for acceptance.